# Novel multiplex-PCR test for *Escherichia coli* detection

Bogumił Zimoń,[1] Michał Psujek,[2] Justyna Matczak,[2] Arkadiusz Guziński,[1] Ewelina Wójcik,[3] Jarosław Dastych[3]

**ABSTRACT** *Escherichia coli* is a diverse and ubiquitous strain of both commensal and pathogenic bacteria. In this study, we propose the use of multiplex polymerase chain reaction (PCR), using amplification of three genes (*cydA, lacY*, and *ydiV*), as a method for determining the affiliation of the tested strains to the *E. coli* species. The novelty of the method lies in the small number of steps needed to perform the diagnosis and, consequently, in the small amount of time needed to obtain it. This method, like any other, has some limitations, but its advantage is fast, cheap, and reliable identification of the presence of *E. coli*. Sequences of the indicated genes from 1,171 complete *E. coli* genomes in the NCBI database were used to prepare the primers. The developed multiplex PCR was tested on 47,370 different *Enterobacteriaceae* genomes using *in silico* PCR. The sensitivity and specificity of the developed test were 95.76% and 99.49%, respectively. Wet laboratory analyses confirmed the high specificity, repeatability, reproducibility, and reliability of the proposed test. Because of the detection of three genes, this method is very cost and labor-effective, yet still highly accurate, specific, and sensitive in comparison to similar methods.

**IMPORTANCE** Detection of *E. coli* from environmental or clinical samples is important due to the common occurrence of this species of bacteria in all human and animal environments. As commonly known, these bacteria strains can be commensal and pathogenic, causing numerous infections of clinical importance, including infections of the digestive system, urinary, respiratory, and even meninges, particularly dangerous for newborns. The developed multiplex polymerase chain reaction test, confirming the presence of *E. coli* in samples, can be used in many laboratories. The test provides new opportunities for quick and cheap analyses, detecting *E. coli* using only three pairs of primers (analysis of the presence of three genes) responsible for metabolism and distinguishing *E. coli* from other pathogens from the Enterobacteriaceae family. Compared to other tests previously described in the literature, our method is characterized by high specificity and sensitivity.

**KEYWORDS** *Escherichia coli*, multiplex PCR, PCR, diagnostics, detection

*E*scherichia coli is a species of a vastly varied nature, and ironically, when it was chosen to be a reference organism for the classification of the *Escherichia* genus in the 19th century (1, 2), microbiologists most likely did not suspect just how diversified organism they started investigating. Since then, *E. coli* has been a model organism used to study bacterial structure, metabolism, and behaviors and was broadly used as a model organism for genetic modifications in laboratory settings (1, 3). In nature, *E. coli* populations have been found both in the aquatic and land environments, as well as, in the gut of vertebrates (4). Not only has it evolved to inhabit the human gut but also is ubiquitously present in habitable-to-human environments and thus is historically connected to us, both as commensal and pathogenic species (3–6).

It is worth noting that the phenotypic classification of *E. coli* strain (pathogenic or non-pathogenic) very often does not align with actual phylogenic affiliation to

Address correspondence to Justyna Matczak, jmatczak@proteonpharma.com.

The authors declare no conflict of interest.

certain clads (4, 7–9). Therefore, genetic investigations looking at the story of *E. coli* divergence from a common ancestor with *Salmonella enterica* within the gastrointestinal (GI) tract very often do not directly translate into their phenotypes, behaviors they depict, virulence, or symptoms they cause (4).

The knowledge as why this species presents both pathogenic and non-pathogenic behaviors is still to be extended. It is suspected that it could be happening due to three, either spontaneous or environmentally induced events: acquisition of new genes; inactivation of antivirulence genes; and point mutations causing change in gene function (4). Some hypothesize that becoming virulent could have been an accidental feature gained along with evolutionarily prevalent traits such as being more persistent or more competitive strain (4, 6, 8).

Complex relations of genomes and phenotypes of strains make it difficult to create a single classification system. Due to practical reasons, many strains have been classified regarding their clinical status, mostly based on phenotypes. Although confusing and difficult to follow, the range of criteria (or their mix) can be used to classify *E. coli* strains, such as pathology it is causing (e.g., diarrhea and intestinal pathogenic *E. coli*—inPEC); infected host (e.g., avian pathogenic *E. coli*—APEC); organ it is affecting (uropathogenic *E. coli*—UPEC); organ and host it is affecting (e.g., cerebrospinal fluid in newborns and newborn meningitis *E. coli*—NMEC), and mode of tissue invasion (e.g., enteroaggregative *E. coli*—EAEC) (3, 4).

Briefly, it could be paraphrased that there is no clear connection between phylogenic affiliation and phenotype, poor systematic differentiation, and a vast spectrum of clinical strains. Along with the rapid inflow of new *in silico* information, researchers often struggle with two issues: current methods require frequent re-evaluation and it is incredibly difficult to design new, reliable, and effective diagnostic tools.

So far, many diagnostic tests have been proposed, including biochemical tests, serotyping, bacteriophage-induced lysis, or different variants of polymerase chain reaction (PCR)-based methods. In many cases, diagnostic approaches utilize more than one kind of test, often relying on biochemical selection by selective growth media to exclude other genera, followed by genetic confirmation of species or even strains.

Various PCR assays have been developed to detect *E. coli* species (10–13). The most common molecular targets include enzymes encoding genes present in the majority of *E. coli* strains (12, 14, 15). Two genes usually chosen for PCR detection are β-D-galactosidase (*lacZ*) and β-D-glucuronidase (*uidA*) (11–14). The utilization of *uidA* can be useful for the classification of *E. coli* lacking enzymatic activity, but possessing the gene sequence (16, 17). However, both *uidA* and *lacZ* were shown to be non-exclusive for *E. coli* (10, 14). In addition, it is troublesome to differentiate enteroinvasive *Escherichia coli* (EIEC) strains from closely related *Shigella*, because bacteria form biochemically and serologically distinct *E. coli* pathovar (18). Both genera are usually distinguished with the use of primers targeting the *lacY* (lactose permease) gene (18, 19), which was shown not unique for *E.coli*. The lactose permease can be found in i.a. *Klebsiella pneumoniae*, *Klebsiella oxytoca*, *Citrobacter freundii,* and *Yersenia Pestis* (20, 21). Moreover, several *lacY* homologs can be found in other *Enterobacteriaceae,* for example, *melY* (melobiose permease) found in *Enterobacter cloacae, rafB* (raffinose transporter) in *Clostridium acetobutilicum* (22).

There have been few single gene-oriented assays developed. One of the early methods was proposed by Candrian (1991) and modified by Wang (1997) and is focused on *malB* gene amplification. The protein encoded by *malB* is called maltoporin and is involved in the transport of maltose (23). Another strategy was described by Walker and involved rtPCR-based detection and enumeration assay targeting a fragment of the *ybbW* gene (211 bp), encoding putative Allantoin transporter (11).

A recent assay developed by Molina (2017) is a multiplex, where *lacZ* and *yaiO* facilitate *E. coli* and coliforms detection. *yaiO* orphan gene fragment amplification in combination with a change of primer hybridization sites toward previously recognized *lacZ* gene (14).

To keep the balance between a relatively low number of primer pairs that target amplicons carry in metabolic and differential information, we propose to use the combination of three genes: *cydA* (part of the cytochrome bd complex), *lacY* (coding for lactose permease), and *ydiV* gene (bacteria motility regulator) as the target for multiplex PCR, which allows to generate three amplicons of specified size indicating that tested strain belongs to *E. coli* species (24–27). Though this publication focuses on PCR-based detection, in the case of environmental samples we would advise initially utilizing microbiological methods (utilization of *E. coli*-specific media such as MacConkey or CHROMagar ECC) to select samples suspected to be *E. coli*. Such a combination of microbiological and genetic methods should grant the best diagnostic results.

## MATERIALS AND METHODS

### Primer design

Primers were designed using 1,171 complete *Escherichia coli* genomes (10.06.2020). Targeted genes [*cydA* (gene encoding Cytochrome bd-I ubiquinol oxidase subunit 1), *lacY* (gene encoding lactose permease), and *ydiV* (gene product participates in quorum sensing and motility regulation)] (25–28) were extracted using NCBI BLAST 2.9.0.+ (28). Multiple sequence alignment was performed in the MAFFT server (03.08.2020: https://mafft.cbrc.jp/alignment/server/), FFT-NS-2 method with sequence direction adjustment. Conserved regions ranging from 17 to 20 bp, containing 40%–55% GC pairs were chosen as primer annealing sites (Table 1). The primers' properties and the products' sizes were selected to enable multiplex PCR to be performed.

TABLE 1 Primer sequences designed in the current study, and previously reported *E. coli* detection assays[a]

| Source | Gene | Primer direction | Amp. Len. (bp) | Primer sequence |
|---|---|---|---|---|
| Current study | *cydA* | Forward | 515 | CGTATGGAGATGGTGAG |
| | | Reverse | | GTAGAACCAGAACGCAGT |
| | *lacY* | Forward | 192 | TTCCCACCGATGCGATT |
| | | Reverse | | GTCACTGTATGTTATTGGCG |
| | *ydiV* | Forward | 330 | CCATTTCTCCAGTGAAGAT |
| | | Reverse | | CCTAACACAAGGGGATAC |
| Walker et al. (11) | *ybbW* | Forward | 211 | TGATTGGCAAAATCTGGCCG |
| | | Reverse | | GAAATCGCCCAAATCGCCAT |
| Molina et al. (10) | *lacZ3* | Forward | 234 | TTGAAAATGGTCTGCTGCTG |
| | | Reverse | | TATTGGCTTCATCCACCACA |
| | *yaiO* | Forward | 115 | TGATTTCCGTGCGTCTGAATG |
| | | Reverse | | ATGCTGCCGTAGCGTGTTTC |
| Horakova et al. (12) | *lacY* | Forward | 463 | ACCAGACCCAGCACCAGATAAG |
| | | Reverse | | GCACCTACGATGTTTTTGACCA |
| | *lacZ* | Forward | 265 | ATGAAAGCTGGCTACAGGAAGGCC |
| | | Reverse | | GGTTTATGCAGCAACGAGACGTCA |
| | *uidA* | Forward | 319 | ATCGGCGAAATTCCATACCTG |
| | | Reverse | | GTTCTGCGACGCTCACACC |
| | *cydA* | Forward | 393 | CCGTATCATGGTGGCGTGTGG |
| | | Reverse | | GCCGGCTGAGTAGTCGTGGAAG |
| Wang et al. (13) | *malB* | Forward | 585 | GACCTCGGTTTAGTTCACAGA |
| | | Reverse | | CACACGCTGACGCTGACCA |
| | *nested-malB* | Forward | 499 | TCTCCTGATGACGCATAGTC |
| | | Reverse | | CACACGCTGACGCTGACCA |

[a]First column indicates the study which yielded *E. coli* detection method presented. Column named "Gene" indicates primer hybridization target, "Amp. Len." stands for amplicon length, expressed in base pairs (bp). The primer sequence can be found in the last column.

## *In silico* PCR

*In silico* PCR (isPCR) was done using perl script (03.08.2020: https://github.com/egonozer/in_silico_pcr), allowing one mismatch. The test was run on a set of 47,370 *Enterobacteriaceae* genomes deposited on NCBI of varying completeness levels (Fig. 1). The list of the strains used to perform the test will be provided upon request. isPCR was carried out on primers designed in the current study and on primers proposed by Horakova (12), Molina (10), Walker (11), and Wang (13), which are summarized in Table 1.

Results of isPCR carried out on primers designed in the current study were analyzed as follows. If all three primer pairs yielded a product of the predicted size, the result was considered positive (predicted as *E. coli* strain). Bacterial genomes yielding amplicons of sizes different from the predicted size or lacking at least one of the products were classified as a negative hit (predicted as non-*E. coli* strain). Results of isPCR carried out on primers designed by others were analyzed according to their criteria.

## Bacteria strains

In all, 20 *E. coli* strains and 20 non-*E. coli* strains belonging to order *Enterobacteriales* were used for laboratory testing and validation of the designed method. *E. coli* strains are as follows: 001PP2015, 002PP2015, 004PP2015, 005PP2015, 007PP2015, 009PP2015, 011PP2015, 012PP2015, 014PP2015, 015PP2015, 016PP2015, 017PP2015, 018PP2015, 032PP2016, 047PP2016, 048PP2016, 049PP2016, 051PP2016, 052PP2016, and 065PP2015. In all, 17 out of 20 *E. coli* used in this study were sequenced, remaining 3 strains (049PP2016, 051PP2016, and 052PP2016) were differentiated molecularly using the melting profile PCR method (29). All utilized *E. coli* strains are genomically different from each other. Strains other than *E. coli* include two *Shigella sonnei* (PCM 1984 and PCM 2336), two *Shigella flexneri* (PCM 89 and PCM 1793), two *Shigella boydii* (PCM 106 and PCM 109), two *Shigella dysenteriae* (PCM 126 and PCM 159), two *Klebsiella pneumoniae* (010PP2016 and 014PP2016), two *Leclercia adecarboxylata* (001PP2018 and 002PP2019), two *Proteus mirabilis* (001PP2016 and 002PP2016), five *Salmonella enterica* (012PP2014, 001PP2016, 065PP2017, 001PP2019, and 002PP2019), and one *Serratia fonticola* (001PP2020). The strains marked as PCM came from the Polish Collection of Microorganisms, the rest are environmental strains from the Proteon Pharmaceuticals Collection. All other than *E. coli* Proteon strains were sequenced.

## Bacterial culture

Each strain used for laboratory testing of the designed method was cultivated separately on lysogenic broth (LB; Biomaxima, PS31, Lublin, Poland) supplemented with agar (Bacteriological LAB-AGAR, AB 03, Lublin, Poland).

The cultures were prepared by streaking frozen stock bacteria strains on LB agar plates. All strains were incubated at 37°C for 24 h ± 2 h. The morphology of bacterial growth has been assessed to preclude possible contaminations.

## DNA isolation

Post-incubation the bacterial biomass has been sampled from 2 to 3 colonies and suspended in 200 µL of PBS (Pol-Aura, Dywity, Poland). After carefully vortexing, bacterial DNA was isolated using the Wizard SV Genomic DNA Purification System (Promega, A2361, Madison, USA). Briefly cell lysis master mix has been prepared by mixing kit with the following proportions: 200 µL Nuclei Lysis Solution; 50 µL of EDTA 0.5 M, pH 8.0; and 5 µL of RNAse and kit-excluded 2 µL of proteinase K (20 mg/mL, activity ≥30 U/mg; A&A Biotechnology, 1019-20, Gdansk, Poland). Isolated DNA has been assessed for concentration and $A_{260/230}$, $A_{260/280}$ ratios using a spectrophotometer (BioPhotometer D30, Eppendorf, Hamburg, Germany) and 1 mm adaptor cuvette (µCuvette G1.0, Eppendorf, Hamburg, Germany) at 260 nm wavelength.

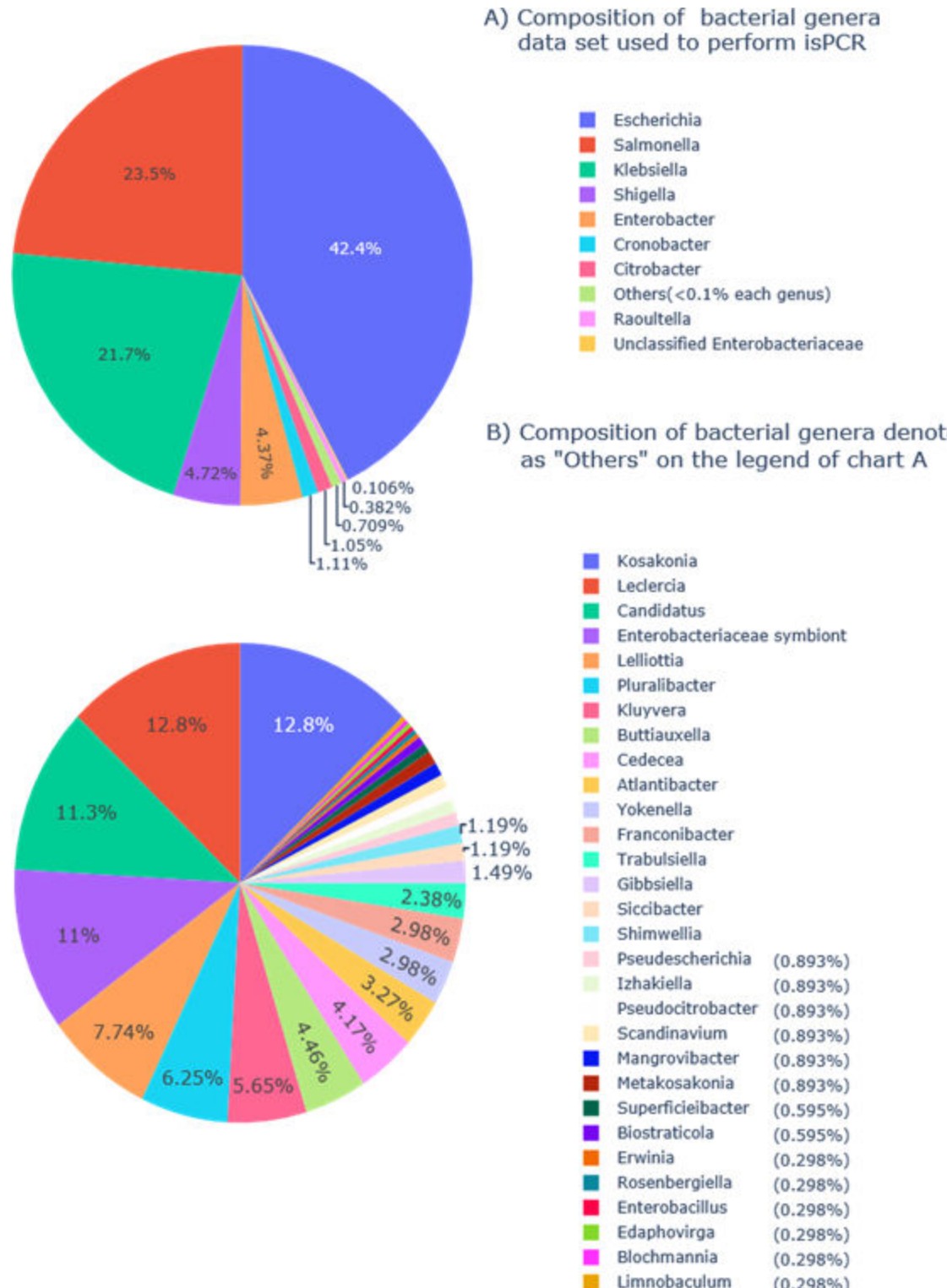

**FIG 1** Pie charts presenting the composition of the Enterobacteriaceae genomes of varying completeness-level data sets used to perform isPCR. (A) Pie chart presenting the contribution of different bacterial genera to 43,730 data set used for isPCR, with 336 genomes denoted as "others" for clarity. (B) Proportion of distinct genera in the "others" data set. Taxa whose percent occurrence could not be clearly denoted on the graph have the value shown on the right side of the legend.

## Multiplex PCR

*cydA, lacY*, and *ydiV* genes (primers in Table 1) were amplified on the matrix of genomic material isolated from strains used for laboratory testing of designed multiplex PCR. The reaction master mix was prepared with the following proportions: 25 µL of Taq PCR Master Mix (EURx, E2520, Gdansk, Poland), 18 µL of PCR-grade water (EURx, E2520, Gdansk, Poland), 1 µL of primers (total 6 µL; Genomed, Warsaw, Poland), and 1 µL of DNA matrix. The nexus GX2 thermal cycler was used (Eppendorf, Hamburg, Germany).

The PCR products underwent electrophoresis on 2% wt/vol agarose gels (EURx, E0301, Gdansk, Poland). The visualization of all three amplified genes was a condition allowing for the diagnosis of *E. coli*, hence two or less products exclude affiliation to *E. coli*. Each gel electrophoresis has been accompanied by a 100 bp DNA marker (Thermo Scientific, SM0321, Vilnius, Lithuania).

## Laboratory validation of designed multiplex PCR

Validation parameters included:

### Sensitivity

The material used to test this parameter constituted DNA isolated from 20 *E. coli* strains. Calculated values fall within 0–1 range, where 1 is the most desirable outcome.

### Specificity

The material used to test this parameter constituted DNA isolated from 20 non-*E. coli* strains belonging to the order *Enterobacterales*. Calculated values fall within 0–1 range, where 1 is the most desirable outcome.

### Repeatability

The material used to calculate repeatability consisted of 10 bacterial strains identified as *E. coli* and 10 non-*E. coli* strains belonging to the order *Enterobacterales*. The measurements ought to be carried out in 10 replicates at once with the use of the same materials and equipment for each sample. Calculated values are within the 0–1 range, where 1 is the most desirable outcome.

### Reproducibility

Material for determination of this parameter consists of five bacterial strains identified as *E. coli* and five non-*E. coli* strains belonging to the order *Enterobacterales*. Calculated values are within the 0–1 range, where 1 is the most desirable outcome.

### Test reliability

The parameter has been measured by changing the manufacturer of the PCR master mix from EURx to Blirt (ReadyMix Taq PCR Reaction Mix, P4600). Material for determination of this parameter consisted of 10 bacterial strains identified as *E. coli* and 10 non-*E. coli* strains belonging to the order *Enterobacterales*. Calculated values are within the 0–1 range, where 1 is the most desirable outcome.

### Analytical sensitivity

The qualitative parameter determines the minimal amount of investigated DNA allowing for the correct interpretation of the result. DNA from a single *E. coli* strain is used to make a series of dilutions for use in PCR. The clear visualization of PCR products by the agarose gel electrophoresis determines the lowest amount needed. To test this parameter, 110, 90, 70, 50, 30, 10, and 2 ng of DNA were added per each reaction.

## RESULTS

### Choice of genes and is PCR

We decided to target three genes: *cydA* (gne encoding Cytochrome bd-I ubiquinol oxidase subunit 1), *lacY* (gene encoding lactose permease), and *ydiV* (gene product participates in quorum sensing and motility regulation) based on the literature. To prepare primers, we obtained sequences of these genes from complete 1,171 *Escherichia coli* genomes deposited in the NCBI database. We performed multiple sequence alignments (MSA), using MAFFT, to choose conservative fragments and based on them we constructed primers for novel multiplex PCR tests. Then, the multiplex PCR was tested on a large data set (47,370 genomes) of diversified *Enterobacteriaceae* genomes (Fig. 1) using isPCR. As a result, we detected 18,963 *E. coli* from 19,802 *E. coli* data set (Fig. 2), revealing 95.76% sensitivity. From 27,568 non-*E.coli* genomes, 27,427 strains remained undetected yielding 99.49% specificity (Fig. 3). Overall accuracy of the novel method reached a value of 97.93% (Fig. 4). The novel method provides high discriminatory power, facilitating resolution between *E. coli* and non-*E.coli*. Especially worth of consideration is the resolution between *E. coli* and *Shigella sp.*, which was really problematic in other diagnostic methods (10–13). For the *Shigella* genus, the test was performed on 2,236 genomes and specificity reached the value of 95.89% (Fig. 5).

Among 839 false-negative (FN) results, 607 are caused by lack of at least one of the products, 225 result from incorrect amplicon size, and 7 hits are caused by both lack of the product and incorrect product size of at least one of the remaining amplicons. In the combined 614 negative results that arose from lack of a product, *lacY* amplicon is most often associated with false-negative hits. It was missing in 390 false negatives, followed by the *ydiV* amplicon missing in 221 cases. The *cydA* amplicon is least associated with the false-negative results—only 38 hits were characterized by a lack of that product.

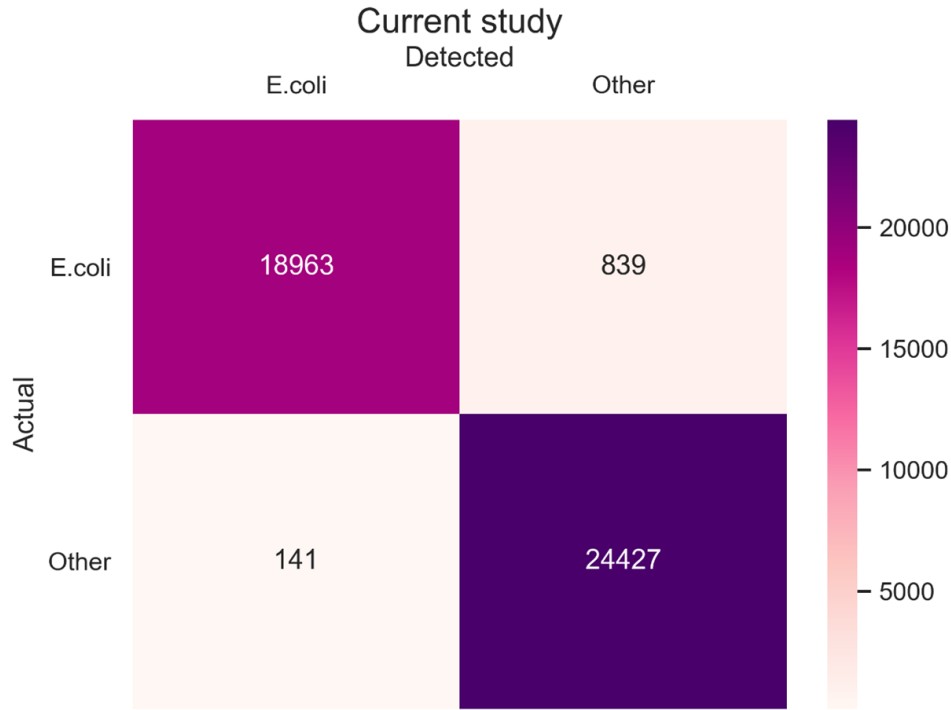

**FIG 2** Confusion matrix presenting proportion of correctly and improperly classified *E. coli* and non-*E. coli* strain (denoted as "Other"). The bar on the right serves as a color scale indicating the number of bacterial genomes—the darker the color the higher the number of genomes.

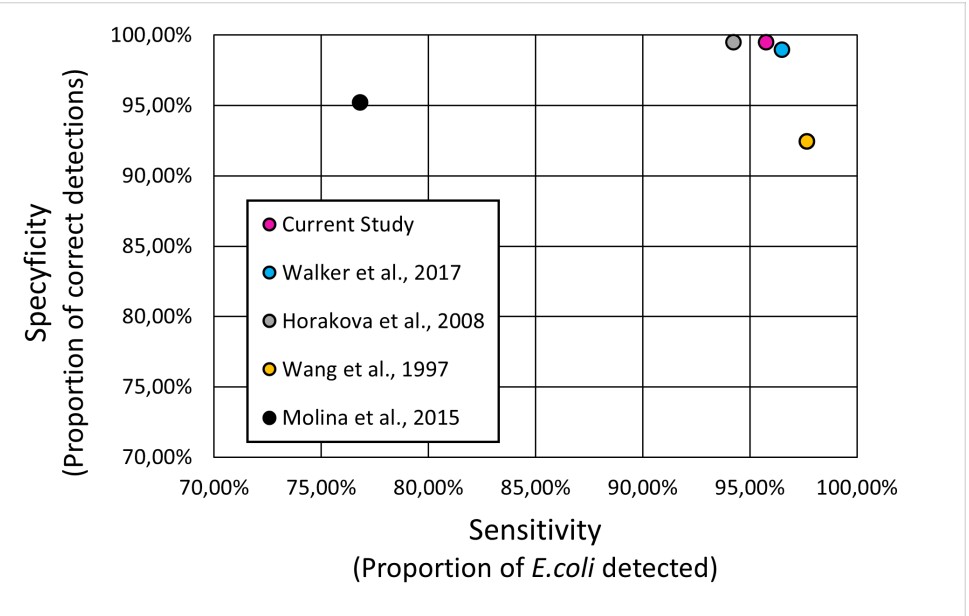

**FIG 3** Scatter plot where x-axis indicates sensitivity/recall (proportion of *E. coli* detected), while the y-axis shows specificity (proportion of correct detections). The closer the dot is to the top rightmost corner the greater the accuracy. The parameters are calculated based on the isPCR performed on the *Enterobacteriaceae* data set, which composition is presented in Fig. 1. The color of the dot fill indicates the detection method tested and corresponds to the legend found in the box. The closer the dot is to the top rightmost corner the greater the accuracy.

There are two combinations most commonly seen in the false-negatives set: *cydA*(+), *lacY*(−), *ydiV*(+), and *cydA*(+), *lacY*(+), *ydiv*(−), where plus (+) stand for amplicon presence and minus (−) amplicon absence—regardless of the amplicon length. The former combination was found in 371 (60.42%) and the latter in 197 (32.08%) undetected *E. coli*. The remaining combinations do not exceed 4% of the false negatives due to lack of product set.

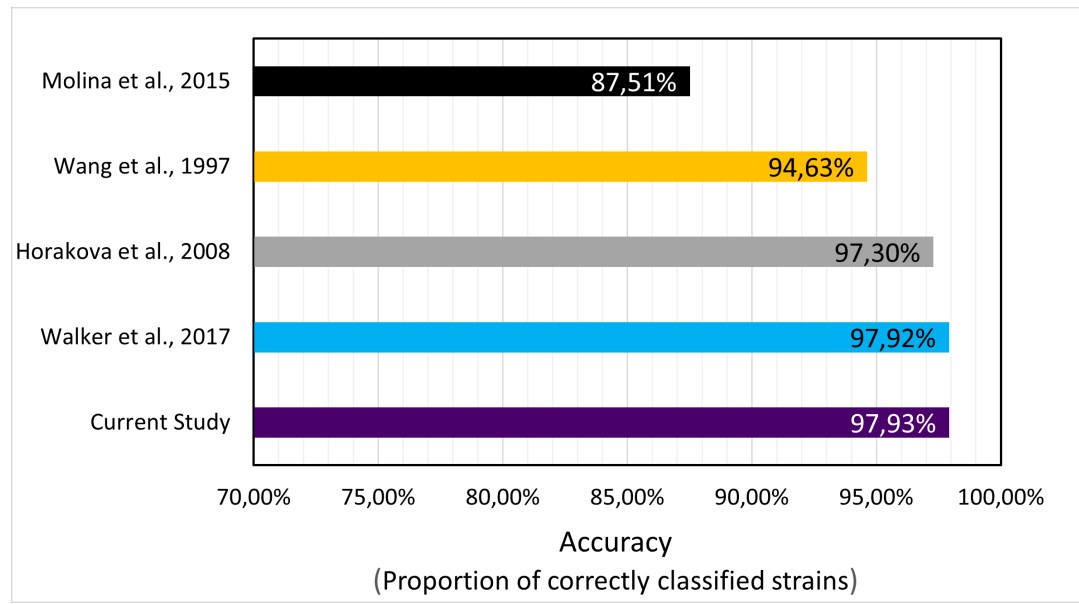

**FIG 4** Bar chart revealing predicted accuracy of different *E. coli* detection methods. The accuracy was calculated based on the isPCR performed on the *Enterobacteriaceae* data set, whose composition is presented in Fig. 1.

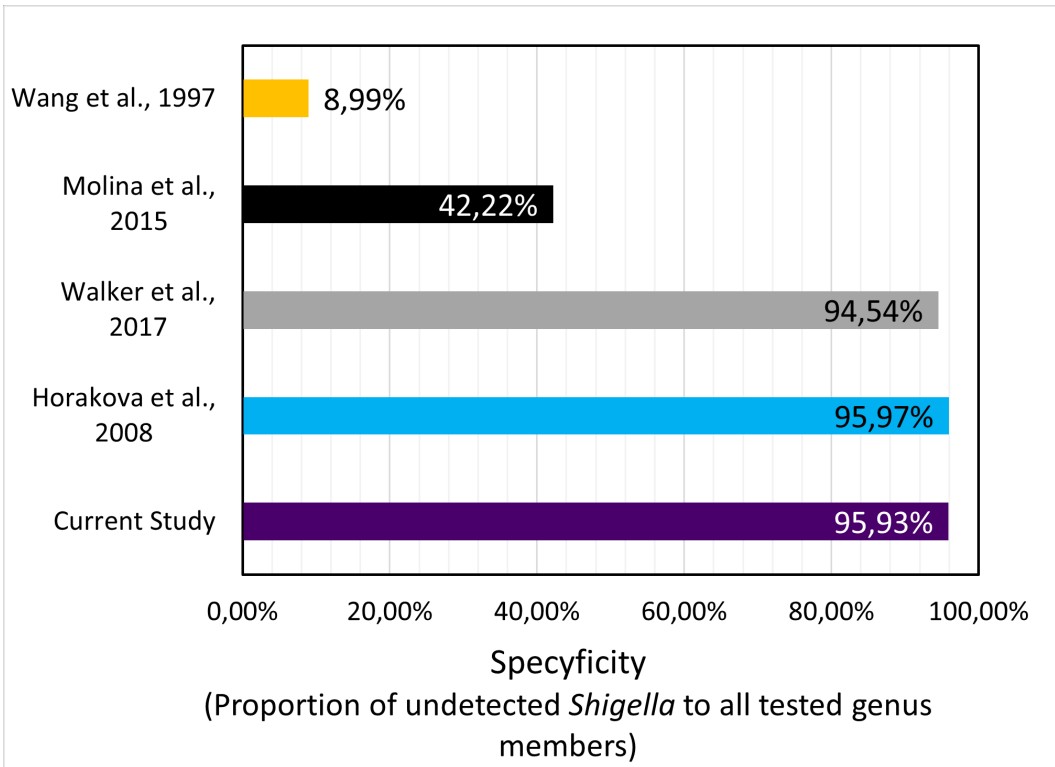

**FIG 5** Bar chart revealing what percentage of *Shigella* genus members remained undetected after subjecting them to primer pairs found in different *E. coli* assays. The results were obtained from isPCR analysis.

There are 236 amplicons with imperfect size belonging to 232 *E. coli* genomes. More than 82% of those products are caused by *ydiV* gene fragment amplification yielding improper length. Incorrect amplification of *lacY* was observed in slightly more than 10% and *cydA* in almost 8% of all amplicons of incorrect length.

Among 141 false-positive (FP) hits out of 27,709 potential targets, almost a third of the set is occupied by genomes belonging to *Escherichia* sp. (40 strains 28.37%). It is possible that part of the unclassified strains actually belongs to *E. coli* species.

The *Shigella* genus is the largest group found in the FP set. There are 91 *Shigella* strains detected, which together stand for 64.54% of false-positive hits. The greatest proportion of the FP set is occupied by *S. sonnei* (60 strains 42.55% of the FP hits), slightly lower by *S. boydii* (22 strains 15.60% of the FP this), *S. flexneri* (7 strains 4.96% of the FP hits), and the lowest by *S. dysenteriae* (2 strains 1.42% of the FP hits). Although several genomes have been improperly classified as *E. coli*, overall specificity with respect to the *Shigella* reached almost 96% (Fig. 5). Except for *S. boydii* (81.67% of strain remained undetected from 120 tested), we can distinguish all remaining species with an average specificity of 97.15% ± 1.20%. The highest specificity considering the Shigella genus was achieved for unclassified *Shigella* sp. (100% tested on 35), slightly lower for *S. flexneri* (98.95% tested on 664 strains), through *S. dysenteriae* (96.97% tested on 66 strains), while second to the lowest specificity belongs to S. sonnei (95.56% tested on 1,351 strains) (Fig. 6).

Remaining false-positive observations include unclassified Enterobacteriaceae (4 strains; 2.84% of the FP observations), two unclassified *Salmonella* sp., and one FP result per each of the listed species: *Salmonella enterica*, one *Enterobacter* sp., one *Escherichia fergusonii,* and one *Klebsiella oxytoca*.

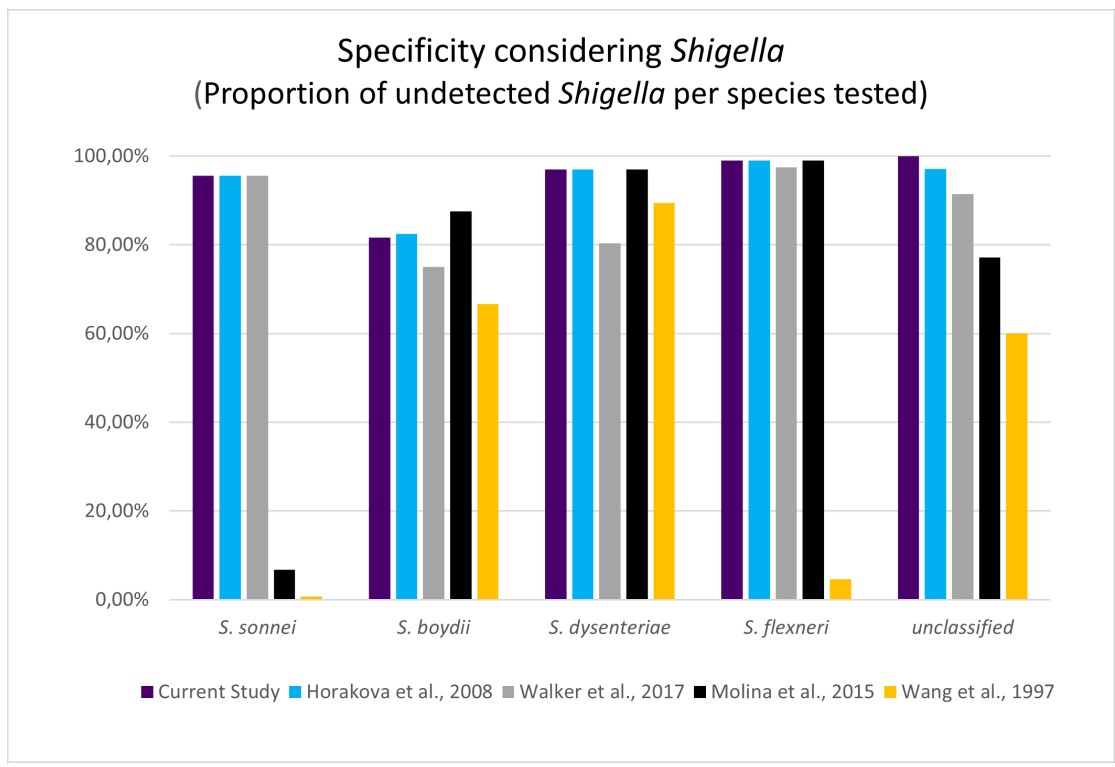

**FIG 6** Column plot indicating what percentage of each of the *Shigella* species remained undetected after subjecting them to different *E. coli* detection assays. The results were obtained from isPCR analysis.

## Comparative analysis of isPCR results

The obtained results were compared to the results of classification methods proposed by Horakova (12), Molina (10), Walker (11), and Wang (13). We performed isPCRs using conditions proposed by them on the same data set and based on sensitivity, specificity, and accuracy compared to the multiplex proposed in this study.

The multiplex PCR developed in the current study shows comparable and even slightly higher accuracy (97.93%) than previously published assays (Fig. 3 and 4). Walker (11) with amplification of a single product shows 97.92% accuracy and, respectively, 98.94% and 96.51% for specificity and sensitivity. Another study utilizing amplification of a single product, Wang (13), shows the same parameters being respectively equal to 94.63%, 92.46%, and 97.66%.

Conversely, Molina (10) proposed aa alternative approach to diagnosis by designing multiplex PCR, which obtained an accuracy of 87.51% with specificity/sensitivity equal to 95.19% and 76.82%. The method proposed by Horakova (12), another example of multiplex PCR, showed results of 97.30% (accuracy), 99.48% (specificity), and 94.25% (sensitivity).

Differentiating *Shigella* from *E. coli* species remains a diagnostic challenge. Our method is characterized by 95.93% specificity toward *Shigella* members. Analogical analysis has been performed for primers designed by Horakova (12), Walker (11), Wang (13), and Molina (10) showing specificity toward *Shigella* sp., respectively, 95.97%, 94.54%, 8.99%, and 42.22%.

Furthermore, we have tested the specificity of current and other methods with respect to each of separate *Shigella* species. The analysis revealed that the current method shows a specificity of 95% or higher toward the detection of *Shigella* species except *S. boydii* (81.67%). These results have been plotted against calculated results from other studies and presented in Fig. 6.

## Wet laboratory PCR validation

The next step was to validate the designed multiplex PCR in laboratory conditions. For this purpose, genomic DNA from 20 *E. coli* and 20 non-*E. coli* strains was isolated. To check the probability of obtaining true-positive (TP) results, genomic material from 20 *E. coli* strains was used. All of the tested specimens have shown amplification of all three genes necessary for the identification of *E. coli* (Fig. 7) giving a 100% probability of obtaining TP result.

To check the probability of obtaining true negative (TN) results, genomic DNA from 20 non-*E. coli* strains from Enterobacterales was used. None of the tested specimens have shown amplification of all three genes necessary for the identification of *E. coli* (Fig. 8) so no false-positive results have been obtained and finally giving a 100% probability of obtaining TN results.

To determine the ratio of the sum of repeatable results obtained for a given sample to the number of samples tested, genomic material from 10 bacterial strains identified as *E. coli* and 10 bacterial strains identified as non-*E. coli* was used. In all, 10 technical replications were performed for each of the strains giving a 100% probability of successfully replicating either TP or TN result.

To check the ratio of the sum of repeatable results obtained for a given sample to the number of samples tested by two different analysts on different days, genomic material from five bacterial strains for each *E. coli* and non-*E. coli* was used. Each of the analysts received a 100% probability of successfully replicating either the TP or TN results.

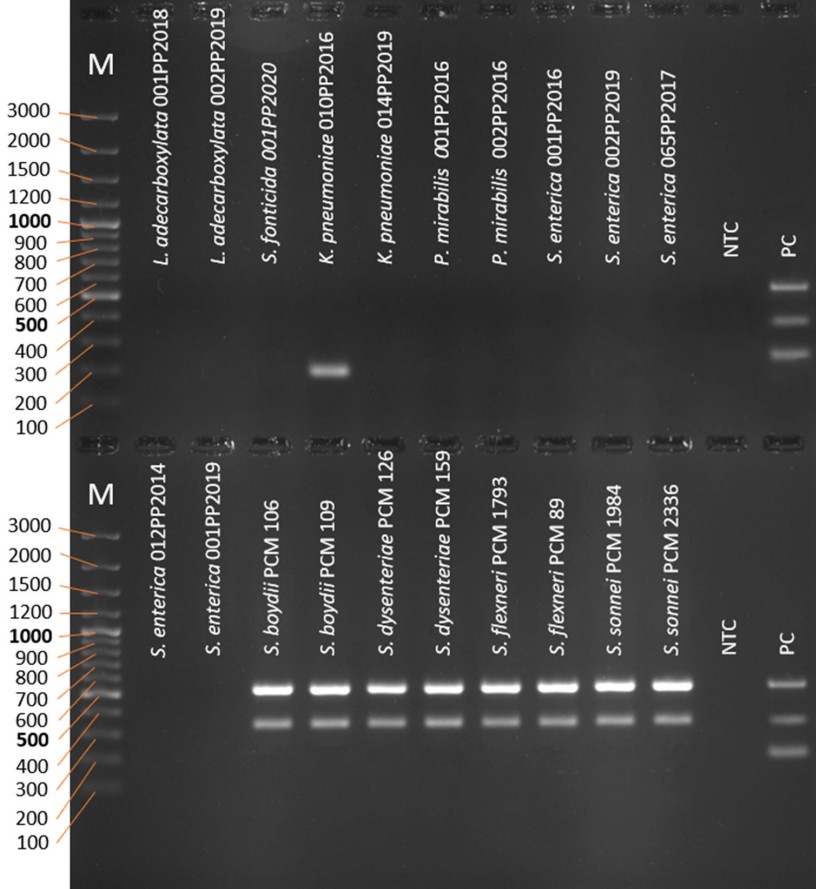

**FIG 7** The PCR products after amplification of cydA, lacY, and ydiV genes as a multiplex on genomic DNA of bacterial strains (electrophoresis in 2% agarose gel). NTC—no template control, PC—positive control. M—marker; the sizes of the molecular size markers are shown in bp on the left side of the figure.

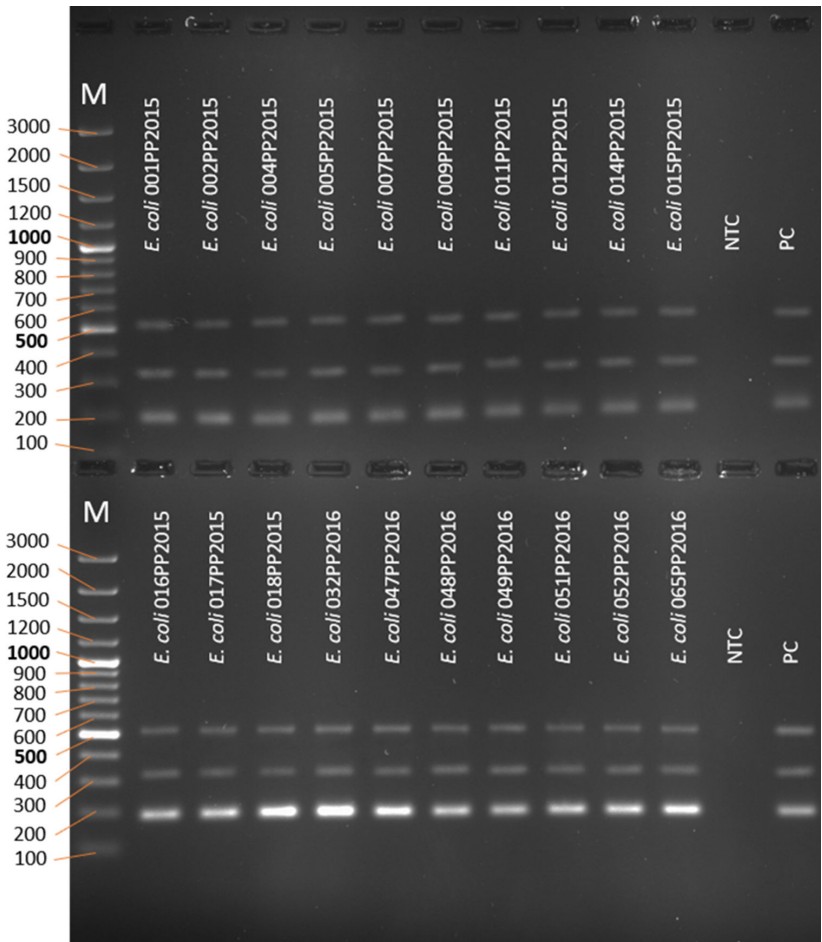

**FIG 8** The PCR products after amplification of cydA, lacY, and ydiV genes as a multiplex on genomic DNA of bacterial strains (electrophoresis in 2% agarose gel). NTC—no template control, PC—positive control. M—marker; the sizes of the molecular size markers are shown in bp on the left side of the figure.

To determine the sensitivity of the test to changes in the reaction environment, including, for example, the use of different equipment, reagents, external conditions, etc., genomic material from 10 *E. coli* and non-*E. coli* strains was used. Two different ready-to-use PCR master mixes have been used: ReadyMix Taq PCR Reaction Mix by Blirt and Taq PCR Master Mix by EURx, giving slight differences in quality of PCR products (visualized on 2% wt/vol agarose gel) but finally there are no diagnostic differences between these two.

Eight different quantities of *E. coli* genomic DNA (110–0 ng/reaction) were used to assess the minimal amount of DNA necessary for yielding the correct diagnostic result. The gel electrophoresis results indicate that for each of the quantities, except 0 ng (no template control—NTC), all three genes necessary for correct diagnosis have been obtained (Fig. 9).

## DISCUSSION

### Overview

The great variety and diversity of clinical *E. coli* pathotypes create a need for precise and fast detection of the *E. coli* species. Multiple genotypic methods have been developed to satisfy this need. However, due to the large number of newly sequenced strains previously established assays require reassessment, and new accurate, low cost, methods have to be introduced. The *E. coli* detection method developed in the current study

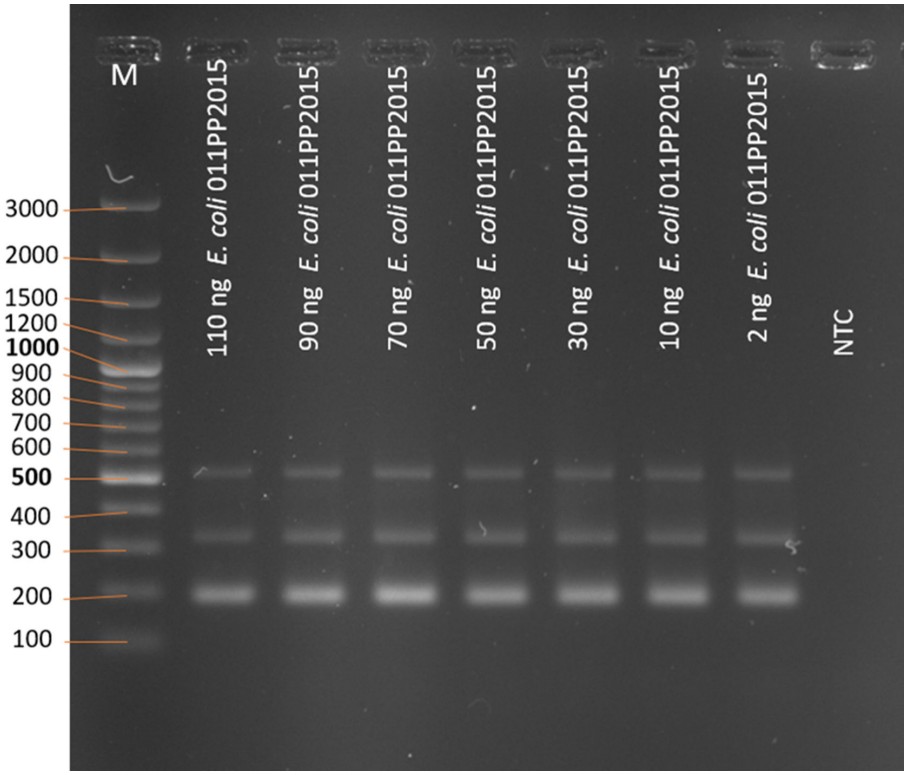

**FIG 9** The PCR products after amplification of cydA, lacY, and ydiV genes as a multiplex on different quantities of genomic DNA of *E. coli* (electrophoresis in 2% agarose gel). NTC—no template control, PC—positive control. M—marker; the sizes of the molecular size markers are shown in bp on the left side of the figure.

shows superb accuracy, attained with the usage of three genes (*cydA*, *lacY*, and *ydiV*), that provide great discriminatory power, initially indicated by *in silico* tests and further confirmed by laboratory PCR detection.

## Metabolic and clinical significance of amplicons

Targeting the aforementioned genes not only ensures competitive accuracy but also provides metabolic insight useful in narrowing down pool of potential microbes present in the sample when *E.coli* is not detected. Primers targeting the *lacY* gene allow differentiation of *E.coli* from *Shigella* sp. and suggest possibility that the tested strain can metabolize lactose, which serves as a useful indicator of coliform presence (30). Utilization of *cydA* was dictated by its previous successful usage in high-specificity *E.coli* detection (12, 31). The gene encodes part of the cytochrome bd—high-affinity oxidase, detecting the amplicon indicates that bacteria of interest retain their anaerobic metabolic functions under low oxygen concentration (32, 33). The protein encoded by the *ydiV* gene was successfully used to distinguish *E. coli* from *E. albertii* and *E. fergusonii*, thus increasing the test specificity (25, 26, 34). Intymin carrying *E. albertii* often undergoes improper classification as enterohemorrhagic *E. coli* (EHEC) or enteropathogenic *E. coli* (EPEC); thus, we believe that usage of this gene as a molecular target is important and justified (35).

## Comparison to other methods

Due to the amplification of three gene fragments in the described detection method, we achieved superior accuracy, attainable when tested *in silico* only by Horakova (12) assay, which however requires an additional primer pair. A limited number of used genes in our detection method makes it cheaper and easier to apply, without sacrificing accuracy

or specificity of detection. We have noted that Wang's (13) classification approach yields greater sensitivity, though an increase in sensitivity led to low specificity and accuracy of that approach. In this instance, it is worth stressing that low specificity suggests that multiple non-*E. coli* species may be classified as such, leading to ambiguous results. A case of the method designed by Molina (10) is slightly different, because though keeping sensitivity at relatively high levels, the sensitivity of the test occurred poorly at only 76.82%, making it at the same time the least accurate method of all (87.51%).

The noteworthy single-gene genotypic method developed by Walker (11) also shows very good specificity, comparable to ours and Horakova's (2008) method. However, when considering overall discriminatory power with respect to *Shigella* species, single-gene assay shows a specificity lower than the multi-gene assay developed in the current study. Wang (13) and Molina (10) also show worse specificity compared to this study, with specificity being 8.99% and 42.22%, respectively. On the other hand, if differentiation of *E. coli* from *Shigellae* is not of utmost concerning, Walker (11) could still be also considered a great environmental detection tool regardless of poorer specificity.

Where all other methods struggled with specificity to *S. boydii*, Molina (10) resulted in the best result of 87.50%. Nonetheless, on other occasions (*S. sonnei, S.dysenteriae, S. flexneri,* and other unclassified *Shigellae*), Horakova (12), Walker (11), and our test presented better results.

The multigene method developed in the current study shows better performance when differentiation from *Shigellae* is crucial and when additional metabolic insight provided by the detection of *cydA*, *lacY,* and *ydiV* genes is beneficial.

It needs to be noted, however, that such *in silico* analyses need practical confirmation. Here we validated the method on a relatively small number of strains, and it would be scientifically correct to increase the number of stains included in wet lab analyses.

## Final remarks

The PCR test described in the current paper should be applied as a final confirmatory tool, on a single colony, and therefore microbiological confirmation is required. When used on a bacterial mixture it is possible to detect one of each amplicon from three different bacterial genomes leading to a false-positive signal. This limitation is usually shared by other multi-gene detection methods; while, single-gene assay, for example, Walker (11) assay is free from it. Although the application of single-gene assays may be convenient, usage of primer pairs targeting multiple genes (*cydA*, *lacY,* and *ydiV*) ensures the lower probability of false-positive detection if the assay is performed according to the protocol, with the advantage of the metabolic insight that each of the detected amplicons carries. The proposed test can be practically used for quick and reliable identification of *E. coli*, for instance in suspected sepsis, infections of the cerebrospinal fluid, and meninges in newborns and others.

## ACKNOWLEDGMENTS

We would like to thank the Polish Collection of Microorganisms for the opportunity to use *Shigella* strains in our analyses.

Research and analysis financed by Proteon Pharmaceuticals.

## AUTHOR AFFILIATIONS

[1]Bioinformatics and Genetics Department, Proteon Pharmaceuticals, Lodz, Lodzkie, Poland

[2]Diagnostics and Monitoring Department, Proteon Pharmaceuticals, Lodz, Lodzkie, Poland

[3]Proteon Pharmaceuticals, Supervisor, Lodz, Lodzkie, Poland

## AUTHOR ORCIDs

Justyna Matczak http://orcid.org/0009-0002-1227-6728

## AUTHOR CONTRIBUTIONS

Bogumił Zimoń, Conceptualization, Investigation, Methodology, Writing – original draft | Michał Psujek, Investigation, Methodology, Validation, Writing – original draft, Writing – review and editing | Justyna Matczak, Conceptualization, Methodology, Writing – original draft, Writing – review and editing | Arkadiusz Guziński, Writing – review and editing | Ewelina Wójcik, Supervision | Jarosław Dastych, Supervision

## ADDITIONAL FILES

The following material is available online.

### Open Peer Review

**PEER REVIEW HISTORY (review-history.pdf).** An accounting of the reviewer comments and feedback.

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
