## [Reviewer comments · Microbiology Spectrum]

Microbiology Spectrum

Novel multiplex-PCR test for *Escherichia coli* detection

Bogumił Zimoń, Michał Psujek, Justyna Matczak, Arkadiusz Guziński, Ewelina Wójcik, and Jarosław Dastych

Corresponding Author(s): Justyna Matczak, Proteon Pharmaceuticals

Review Timeline:

Submission Date:	November 6, 2023
Editorial Decision:	February 10, 2024
Revision Received:	March 13, 2024
Accepted:	March 27, 2024

Editor: Bernadette Connors

Reviewer(s): Disclosure of reviewer identity is with reference to reviewer comments included in decision letter(s). The following individuals involved in review of your submission have agreed to reveal their identity: Wioletta Adamus-Białek (Reviewer #1); Edwin Barrios-Villa (Reviewer #3)

Transaction Report:

DOI: <https://doi.org/10.1128/spectrum.03773-23>

Re: Spectrum03773-23 (Novel multiplex-PCR test for Escherichia coli detection)

Dear Dr. Justyna Katarzyna Matczak:

Thank you for the privilege of reviewing your work. Below you will find my comments, instructions from the Spectrum editorial office, and the reviewer comments.

Revision Guidelines

Sincerely,
Bernadette Connors
Editor
Microbiology Spectrum

Reviewer #1 (Comments for the Author):

Dear Editor-in-Chief,
Dear Authors,

The developed test seems to be a very promising method for the molecular detection of E. coli. The results were described very clearly, but there are some doubts.

Detailed comments below:

1. Line 25-28: The aim of the project is not to distinguish pathogenic from non-pathogenic *E. coli*, but to identify them regardless of pathogenicity. The introduction in the abstract should be focused on the quick identification of *E. coli* and why it is needed (role in various infections, septic potential).
2. Was the strain classification carried out in accordance with applicable standards? If yes - please specify, if no - then use the name "isolates" because we don't know if these are the same strains or different ones. In this case, laboratory validation seems questionable, different strains of *E. coli* should be used, and the study should be carried out on a larger group size.
3. „starters" please change to „primers"
4. Why does in silico analysis include a narrower group of species than in vitro analysis? There should be the same taxonomic levels. The test examined only two species belonging to families other than Enterobacteriaceae, so it is difficult to conclude that the in vitro analysis covers validation of the method for the Enterobacterales order. Please clarify this, it seems that the method is more validated for Enterobacteriaceae.
5. It would be better to change *Proteus* sp. and *Serratia* sp. to *Enterobacter* sp. and *Citrobacter* sp. in wet lab PCR validation.
6. Fig. 2 the statistical analysis should be performed whether false results (differences) are significant. If the differences are significant, it will strengthen the reliability of the in silico test, which does not change the fact that false results should be expected in a given percentage of cases.
7. Please detail the limitations of the work and present a proposal for the practical use of the test. It seems that the method may be useful when there is a need for quick identification of *E. coli* (suspected epidemic, sepsis), but phenotypic/microbiological confirmation of the sample is necessary.

Reviewer #3 (Comments for the Author):

Zimon et al., presented a manuscript entitled "Novel multiplex-PCR test for *Escherichia coli* detection". They reviewed a huge amount of publicly available genomic data in databases to determine the presence or absence of amplicons by in silico PCR. The authors also compared using the same strategy other PCR-based methods previously described and proposed a set of primers to amplify *cydA*, *lacY* and *ydiV* genes. A comprehensive pipeline is also proposed for results interpretation and avoiding false-positive -negative results.

My observations or suggestions are as follows:

- L93. Please erase the blank space after de β for the *lacZ* protein.
- L106. Why *molB* is underlined?
- Methodology: When referring to reagents or equipment, please indicate (Brand name, city, country) if applicable.
- L237 You mentioned the finding of imperfect size, do you analyzed the amplicon obtained? if it was isPCR, then you have access to de genome sequence, it may be worth it to analyze if there is a variant or an allele of the gene, it would even be interesting to see if it is a gene that is in other enterobacteria not sought after that could give a cross-reaction.
- What happened with the unclassified genomes? It would be interesting to inquire deeper, particularly if the genome sequences are available, an ANI or even a BLAST could give you a good sight.
- Gel figures. Please consider to invert the colour, the bands might be more visible.
- There is another reverse primer designed by Carreon-Leon to improve the Walker PCR, it worth to be tested with the proposed (<https://repositorioinstitucional.buap.mx/items/cd2e0592-f469-4931-9c13-99f11082a554>)

Dear Editor-in-Chief,

Dear Authors,

The developed test seems to be a very promising method for the molecular detection of *E. coli*. The results were described very clearly, but there are some doubts.

Detailed comments below:

1. Line 25-28: The aim of the project is not to distinguish pathogenic from non-pathogenic *E. coli*, but to identify them regardless of pathogenicity. The introduction in the abstract should be focused on the quick identification of *E. coli* and why it is needed (role in various infections, septic potential).
2. Was the strain classification carried out in accordance with applicable standards? If yes - please specify, if no - then use the name "isolates" because we don't know if these are the same strains or different ones. In this case, laboratory validation seems questionable, different strains of *E. coli* should be used, and the study should be carried out on a larger group size.
3. „starters“ please change to „primers“
4. Why does *in silico* analysis include a narrower group of species than *in vitro* analysis? There should be the same taxonomic levels. The test examined only two species belonging to families other than *Enterobacteriaceae*, so it is difficult to conclude that the *in vitro* analysis covers validation of the method for the *Enterobacterales* order. Please clarify this, it seems that the method is more validated for *Enterobacteriaceae*.
5. It would be better to change *Proteus* sp. and *Serratia* sp. to *Enterobacter* sp. and *Citrobacter* sp. in wet lab PCR validation.
6. Fig. 2 the statistical analysis should be performed whether false results (differences) are significant. If the differences are significant, it will strengthen the reliability of the *in silico* test, which does not change the fact that false results should be expected in a given percentage of cases.
7. Please detail the limitations of the work and present a proposal for the practical use of the test. It seems that the method may be useful when there is a need for quick identification of *E. coli* (suspected epidemic, sepsis), but phenotypic/microbiological confirmation of the sample is necessary.

- 1. Line 25-28: The aim of the project is not to distinguish pathogenic from non-pathogenic *E. coli*, but to identify them regardless of pathogenicity. The introduction in the abstract should be focused on the quick identification of *E. coli* and why it is needed (role in various infections, septic potential).**

This is a valid point, the test does not allow the distinction between pathogenic and non-pathogenic *E. coli*, but it is a quick method of detecting *Escherichia coli*, which has great potential for use in suspected sepsis, epidemics, etc. The abstract has been changed to focus on indicated comment.

- 2. Was the strain classification carried out in accordance with applicable standards? If yes - please specify, if no - then use the name "isolates" because we don't know if these are the same strains or different ones. In this case, laboratory validation seems questionable, different strains of *E. coli* should be used, and the study should be carried out on a larger group size.**

17 out of 20 *E. coli* used in this study were sequenced. According to analysis they are different from each other, hence use of term "strain" is justifiable. Their sequences will be deposited to NCBI database soon. Remaining 3 *E. coli* were differentiated molecularly using melting profile PCR (PCR-MP) method. This information will be added to the text.

We could agree for additional confirmation of validation in the future, based on a bigger group of strains.

- 3. „starters" please change to „primers"**

Agree –done.

- 4. Why does *in silico* analysis include a narrower group of species than *in vitro* analysis? There should be the same taxonomic levels. The test examined only two species belonging to families other than Enterobacteriaceae, so it is difficult to conclude that the *in vitro* analysis covers validation of the method for the Enterobacterales order. Please clarify this, it seems that the method is more validated for Enterobacteriaceae.**

The *in silico* analysis having access to large quantities of sequences focused on distinguishing *E. coli* from bacterial species belonging to order of extreme similarity mainly Enterobacteriaceae. The combination of genes and primer annealing sites selected is not only highly specific for Enterobacteriaceae but directly to *E. coli*. Those genes may often be absent in other bacterial families or the gene homologues so distant that successful primer annealing and amplification is unlikely. Knowing that the test shows great accuracy in the narrow but most difficult to distinguish family, it can be inferred that the test performance will be equal or even higher when applied to distant bacterial families. Although generation of product combinations with appropriate size is unlikely in other bacterial families, an *in vitro* test was performed in an extended set of bacterial

taxa in order to prove beyond reasonable doubt that this is the case. Due to lack of access to a greater number of *Enterobacteriaceae* species the *in vitro* test focused on selecting highly similar species from that family mainly *Shigella* and *Escherichia*. This combination of taxonomically broader study *in vitro* (but limited in number of species tested) combined with narrower taxonomically but more thorough (greater number of tested bacterial species) *in silico* test should accurately cover the complexity of *E. coli* detection.

5. It would be better to change *Proteus* sp. and *Serratia* sp. to *Enterobacter* sp. and *Citrobacter* sp. in wet lab PCR validation.

Perhaps yes, we could agree to that, saying that the group of species could be wider and include *Enterobacter* and *Citrobacter*. However, there were none available to us during validation, and it is a limitation, yet checking two additional species is still beneficial due to increase in variety.

6. Fig. 2 the statistical analysis should be performed whether false results (differences) are significant. If the differences are significant, it will strengthen the reliability of the *in silico* test, which does not change the fact that false results should be expected in a given percentage of cases.

In order to establish statistical significance of the binary classification with certain class imbalance, we have performed a chi-squared test. This test revealed chi-squared value of 40617 with p-value lower than $2.225 * 10^{-308}$ (more precise result could not be obtained due to python floating point limit). The classification results are highly different from what would be expected from random distribution of classes, indicating that the test efficiently distinguishes *E.coli* from other bacterial species. This finding was added to the main text of the publication.

7. Please detail the limitations of the work and present a proposal for the practical use of the test. It seems that the method may be useful when there is a need for quick identification of *E. coli* (suspected epidemic, sepsis), but phenotypic/microbiological confirmation of the sample is necessary.

The limitation of relatively small number of strains used for *in vitro* validation was added in text (L359-361). Furthermore, remarks to previously included limitations were added to stress these to the reader. Generally limitations have not been omitted – e.g. final remarks.

Proposal for the practical use of the test was added in text.

8. L93. Please erase the blank space after de β for the *lacZ* protein.

Agree- done.

9. L106. Why *molB* is underlined?

Agreed- changed, it was editing error.

10. Methodology: When referring to reagents or equipment, please indicate (Brand name, city, country) if applicable.

Agree- done.

- 11. L237 You mentioned the finding of imperfect size, do you analyzed the amplicon obtained? if it was isPCR, then you have acces to de genome sequence, it may be worth it to analyze if there is a variant or an allele of the gene, it would even be interesting to see if it is a gene that is in other enterobacteria not sought after that could give a corss-reaction.**

While it is interesting from a functional and evolutionary standpoint to look into differences in amplicon sizes, that could be due to various gene alleles present, it does not diminish predictive capabilities of the test significantly. It is equally likely that due to incomplete sequence of bacterial genomes annealing site could be at the contig end, which is a technical trait of incomplete sequencing. It is outside of the current study scope to perform this analysis, however in the future studies it could be an interesting consideration. We acknowledge that there is potential of cross-reactivity revealed by both in silico and in vitro studies, which is alleviated by utilizing selective medium as a prerequisite to genetic test.

- 12. What appened with the unclassified genosmes? It would be interesting to inquire deeper, particularly if the genome sequences are available, an ANI or even a BLAST could give you a good sight.**

It is a very interesting consideration, several of the unclassified *Escherichia* species were indeed compared through BLAST search to other bacterial genome sequences. This search usually results in finding high similarity to other unclassified *Escherichia* genomes, with no significant similarity to specific species. From the perspective of the current study, the species containing required PCR products found may indeed represent *E. coli*. However, due to lack of additional evolutionary evidence and access to unclassified bacterial isolates it is not possible for the authors to ascertain this with full confidence.

- 13. Gel figures. Please consider to invert the colour, the bands might be more visible.**

While this is a good observation, authors do not see much need for inversion of colours given very good clarity of original images. Them being as little edited as possible makes them more reliable, and improves time-efficiency during writing process.

- 14. There is another reverse primer designed by Carreon-Leon to improve the Walker PCR, it worth to be tested with the proposed (<https://repositorioinstitucional.buap.mx/items/cd2e0592-f469-4931-9c13-99f11082a554>)**

While it is great to see improvement over the Walker approach the test is highly similar to the already evaluated ybbW amplification in the current study targeting the same gene. One could expect similar advantages of the test, mainly increased sensitivity due to use of a single product, but diminished specificity, because only one metabolic and evolutionary trait is used in the

comparison. The authors discussed and evaluated those advantages and limitations of single vs multiple PCR products in the current study.

Re: Spectrum03773-23R1 (Novel multiplex-PCR test for Escherichia coli detection)

Dear Dr. Justyna Katarzyna Matczak:

Your manuscript has been accepted, and I am forwarding it to the ASM production staff for publication. Your paper will first be checked to make sure all elements meet the technical requirements. ASM staff will contact you if anything needs to be revised before copyediting and production can begin. Otherwise, you will be notified when your proofs are ready to be viewed.

Sincerely,
Bernadette Connors
Editor
Microbiology Spectrum

Reviewer #1 (Comments for the Author):

The work has been corrected in accordance with the comments and all doubts have been fully clarified.

Reviewer #3 (Comments for the Author):

The authors addressed all the comments/suggestions. I have no further commentaries